# Pregnant Women at Low Risk of Having a Child with Fetal and Neonatal Alloimmune Thrombocytopenia Do Not Require Treatment with Intravenous Immunoglobulin

**DOI:** 10.3390/jcm12175492

**Published:** 2023-08-24

**Authors:** Jens Kjeldsen-Kragh, Gregor Bein, Heidi Tiller

**Affiliations:** 1Department of Clinical Immunology and Transfusion Medicine, University and Regional Laboratories, Akutgatan 8, 221 85 Lund, Sweden; 2Department of Laboratory Medicine, University Hospital of North Norway, 9019 Tromsø, Norway; 3Institute for Clinical Immunology, Transfusion Medicine and Hemostasis, Justus-Liebig-University, 35392 Giessen, Germany; gregor.bein@immunologie.med.uni-giessen.de; 4German Center for Feto-Maternal Incompatibility, University Hospital Giessen and Marburg, Campus Giessen, 35392 Giessen, Germany; 5Department of Obstetrics and Gynecology, University Hospital of North Norway, 9019 Tromsø, Norway; heidi.tiller@uit.no; 6Women’s Health and Perinatology Research Group, Department of Clinical Medicine, UiT The Arctic University of Norway, 9019 Tromsø, Norway

**Keywords:** pregnancy, alloimmunization, intracranial hemorrhage, intravenous immunoglobulin, platelet antibodies

## Abstract

Fetal and neonatal alloimmune thrombocytopenia (FNAIT) is a rare condition in which maternal alloantibodies to fetal platelets cause fetal thrombocytopenia that may lead to intracranial hemorrhage (ICH). Off-label intravenous immunoglobulin (IVIg) has for 30 years been the standard of care for pregnant women who previously have had a child with FNAIT. The efficacy of this treatment has never been tested in a placebo-controlled clinical trial. Although IVIg treatment may improve the neonatal outcome in women who previously have had a child with FNAIT-associated ICH, the question is whether IVIg is necessary for all immunized pregnant women at risk of having a child with FNAIT. The results from some recent publications suggest that antenatal IVIg treatment is not necessary for women who are (1) HPA-1a-immunized and HLA-DRB3*01:01-negative, (2) HPA-1a-immunized with a previous child with FNAIT but without ICH or (3) HPA-5b-immunized. If IVIg is not used for these categories of pregnant women, the amount of IVIg used in pregnant women with platelet antibodies would be reduced to less than ¼ of today’s use. This is important because IVIg is a scarce resource, and the collection of plasma for the treatment of one pregnant woman is not only extremely expensive but also requires tremendous donor efforts.

## 1. Introduction

Fetal and neonatal alloimmune thrombocytopenia (FNAIT) is a rare but potentially severe fetal–maternal condition in which maternal alloantibodies to paternally inherited platelet antigens cause thrombocytopenia in the fetus/newborn. Thus, there is resemblance to the pathogenesis of hemolytic disease of the fetus and newborn (HDFN) where an RhD-negative mother can be RhD-immunized if she gives birth to an RhD-positive child. FNAIT, however, differs from HDFN, as immunization against paternally inherited platelet antigens more often occurs in the first incompatible pregnancy and severe clinical outcomes are seen even in firstborns [1].

The true Incidence of FNAIT is not known but has been estimated at around 1 in 1500 pregnancies [1]. The clinical spectrum varies from mild thrombocytopenia to severe intracranial hemorrhage (ICH), which has been estimated to occur in around 1 in 10,000 unselected pregnancies [2]. In Caucasians, antibodies to human platelet antigen (HPA)-1a account for approximately 80% of FNAIT cases [1] and HPA-5b antibodies have been considered to be implicated in around 15% of the cases [3]. In some Asian populations, however, the majority of antibodies detected in suspected FNAIT cases are HPA-5b antibodies followed by HPA-4b antibodies [4].

For more than 30 years, it has been known that the propensity to develop antibodies against HPA-1a is closely associated with HLA-DRB3*01:01 [5]. Recently it was shown that the risk of having a neonate with severe FNAIT is extremely low for an HPA-1a-immunized, HLA-DRB3*01:01-negative mother [6].

The HPA-1a/b polymorphism is located on the β3 integrin chain of the αIIbβ3 complex that constitutes the platelets’ fibrinogen receptor. As the β3 chain forms heterodimers with the αV chain of the endothelial cells’ vitronectin receptor, HPA-1a antibodies also bind to endothelial cells. This may have pathophysiological importance as HPA-1a antibodies that are specific for the αVβ3 complex are suggested to play a key role in women who have given birth to a child with ICH [7]. However, it is not fully understood if or how such subgroups of HPA-1a antibodies relate to ICH risk, and the clinical use of such antibody analyses has not been tested in prospective studies. Whether the quantification of maternal HPA-1a antibody levels could be useful to predict the risk of ICH is also not known. Therefore, based on current knowledge, the only known ICH risk predictor is an obstetric history with FNAIT.

According to early reports, ICH occurs in 10–30% of FNAIT cases [8,9,10,11]. Moreover, as the recurrence rate was reported to be very high in subsequent pregnancies and the severity was believed to increase compared to that of the previous affected fetus/infant, similar to in HDFN [8,12], the need for fetal bleeding prophylaxis became obvious. However, due to the rarity of ICH in FNAIT cases, there are many uncertainties regarding the natural history of FNAIT. For this reason, the scientific community has relied heavily on data from retrospective case series collected by fetal–maternal medicine specialists, neonatologists and reference laboratories spanning two or more decades. This implies fragmentary clinical data, a lack of appropriate control groups and reduced antibody avidity due to repeated thaw and freeze cycles of historic sera.

In this review, we will critically examine the scientific foundation for today’s management of FNAIT, and we will suggest that the majority of pregnant women at risk of having a child with FNAIT may not necessarily need the currently applied treatment with intravenous immunoglobulin (IVIg).

## 2. Intravenous Immunoglobulin Is the Predominant Treatment for Platelet-Immunized Women

In a landmark paper by Bussel and co-workers from 1988 [13], they reported the results of IVIg treatment (IVIg 1 g/kg/week) in seven pregnant women who previously had had a child with FNAIT. In all children, the platelet count was higher than in their older FNAIT-affected sibling. This paper was subsequently followed by a large number of reports where IVIg, with or without corticosteroids, was used for pregnant women at risk of FNAIT in their newborn. Many of these studies were case reports and case series, while some were randomized clinical trials (RCT) where IVIg was tested against IVIg plus corticosteroids or where different doses of IVIg were compared; for an overview, see Rayment et al., 2011 [14] or Winkelhorst et al., 2017 [15].

Surprisingly, despite decades of usage of IVIg in platelet-immunized women, it is still unknown how IVIg works. Several potential mechanisms of IVIg in platelet-immunized pregnant women were recently summarized by Wabnitz and co-workers [16].

Due to the rarity of FNAIT, it is a considerable challenge to conduct an RCT. Unfortunately, a placebo-controlled RCT demonstrating the efficacy of IVIg treatment for risk pregnancies has never been conducted. Consequently, the off-label usage of IVIg has become the standard treatment for pregnant women who previously have had a child with FNAIT. This treatment modality has become so rooted that it is integrated into several clinical guidelines for the management of risk pregnancies [17,18,19]. Today, it has even been considered unethical to conduct a placebo-controlled RCT to demonstrate IVIg’s efficacy [14].

Although IVIg treatment is usually considered safe, significant side effects have been reported, such as aseptic meningitis, renal failure, hemolytic anemia and thrombotic complications [20,21]. More than 80% of women experience headache during treatment, which has a non-negligible adverse impact on life quality [20].

## 3. The Traditional View on the Severity of FNAIT in Subsequent Pregnancies

Due to the high recurrence rate of severe FNAIT, the efficacy of IVIg has been evaluated by comparing the neonatal outcome in an index pregnancy with the outcome of a subsequent IVIg-treated pregnancy. These studies have indicated that IVIg treatment of women with platelet antibodies is efficient in preventing fetal/neonatal ICH; for details, see Winkelhorst et al., 2017 [15]. Thus, the reported efficacy of IVIg relies on the assumption that the neonatal outcome becomes worse in subsequent incompatible pregnancies, similar to in HDFN. This assumption is based on a frequently referenced study by Radder et al. [22]. In this study, the risk of ICH was calculated to be greater than 70% if the mother previously had had a child with ICH, and 7% if the previous FNAIT-affected sibling did not suffer with ICH. However, these calculations were based on retrospective data from case reports and smaller observational series where samples from the mother and child had been sent to a platelet immunology laboratory on the suspicion of FNAIT. For such retrospective studies, there is a potential risk of publication bias—cases where a woman has given birth to two or more children with FNAIT-associated ICH are more likely to be published than cases where the neonatal outcome of a subsequent pregnancy had a more benign course than the previous. Thus, there is considerable uncertainty regarding this estimate of the ICH recurrence rate and severity of FNAIT in subsequent pregnancies. Ideally, this should be evaluated in a population of pregnant women not treated with IVIg, where the risk of selection bias is minimal.

## 4. Identification of Less Severe Courses of FNAIT

The view that FNAIT becomes more severe in subsequent pregnancies has been challenged by observations from Norway. Prospective data have shown that the levels of HPA-1a antibodies in most multigravida women declined during pregnancy [23], and that the neonatal platelet count in 23 of subsequent HPA-1a-immunized pregnancies increase or remain unchanged [24]. Furthermore, as IVIg only has a very small role in the Norwegian FNAIT management strategy, the neonatal outcome of Norwegian women with platelet antibodies represents the best available data on the natural history of FNAIT. Assuming that IVIg is efficient in preventing anti-HPA-1a-associated ICH, one would expect the neonatal outcome of platelet-immunized pregnancies in Norway to be worse than in other countries where IVIg is used for this category of patients. This question was addressed in a recent study [25] where the neonatal outcome of platelet-immunized women not treated with IVIg was compared with the outcome of women who received IVIg during pregnancy [15]. The pregnant women were stratified according to their previous obstetric history. Women with a prior child with anti-HPA-1a-associated ICH were categorized as “high-risk” pregnancies, whereas women with a prior child with FNAIT without ICH were categorized as “low-risk” pregnancies. In the group of high-risk pregnancies, there were five cases of fetal/neonatal ICH among ninety IVIg-treated pregnancies, equivalent to 5.6% (95% confidence interval (CI): 2.4–12.4%), and two cases among seven children (29%, 95% CI: 8.2–64.1%) from non-IVIg-treated pregnancies (*p* = 0.08). Among the low-risk pregnancies, 2 of 313 IVIg-treated women gave birth to children with ICH (0.6%, 95% CI: 0.2–2.3%), as opposed to 0 cases (0.0%, 95% CI: 0.0–5.7%) among 64 children from non-IVIg-treated pregnancies (*p* = 1.00) [25]. This study is considered to provide the hitherto most reliable data for the evaluation of IVIg treatment in low-risk pregnancies. As the neonatal outcome of low-risk pregnancies in Norway did not seem to be less favorable than in the IVIg-treated control group [15], it seems reasonable to question if IVIg treatment is really necessary for all HPA-1a-immunized pregnant women who previously have had a child with FNAIT but without ICH. However, it is important to emphasize that non-IVIg treatment in Norway is not the same as no intervention. When the risk of FNAIT is recognized before birth, several measures are taken according to Norwegian clinical guidelines, including delivery by caesarean section 1–2 weeks prior to term if the maternal HPA-1a antibody concentration is >3 IU/mL as well as prompt transfusion with compatible platelets to the newborn; for details, see Tiller et al., 2020 [26].

Antibodies against HPA-5b have for years been considered the second most common platelet antibody responsible for FNAIT. This notion was challenged by Alm and co-workers in a recent study from Gießen, Germany [3]. Neither data retrieved from the literature, nor retrospective data from 761 pairs of maternal/fetal samples from the platelet immunology laboratory in Gießen, supported the hypothesis that HPA-5b antibodies cause severe thrombocytopenia or bleeding complications in the fetus/newborn [3].

The HPA-5b antigen is far more immunogenic than, for instance, the HPA-1a antigen, and consequently, around 2% of pregnant women are HPA-5b-immunized as compared to only 0.2% who are HPA-1a-immunized [3]. Thus, it is questionable if HPA-5b antibodies can cause fetal/neonatal thrombocytopenia, or whether the presence of maternal HPA-5b antibodies in a thrombocytopenic newborn is merely coincidental [3].

In a recent Dutch cohort study [27], there were four cases with severe fetal/neonatal bleeding among 40 HPA-5b-immunized women with an incompatible fetus. Interestingly, they also reported [27] the neonatal outcomes of eight HPA-5b-negative women with high levels of HPA-5b antibodies where there was no HPA-5b incompatibility between the mother and fetus. Of these eight children, there was one child with ICH and one child with thrombocytopenia and skin bleeding; none of these children suffered from any other clinical condition known to be associated with thrombocytopenia or ICH [27]. Hence, the incidence of severe bleeding, including ICH, among HPA-5b-incompatible immunized pregnancies (4 of 40) was not higher than among HPA-5b compatible pregnancies (1 of 8). Furthermore, the prevalence of maternal HPA-5b antibodies in pregnancies from 105 neonates with ICH (1.9% [28]) was not different from the prevalence of maternal HPA-5b antibodies in the healthy controls (1.96% [3]), calling into question the causal role of HPA-5b antibodies in ICH.

Despite the fact that up to 2% of all women who have been pregnant are HPA-5b-immunized, we have only been able to find one single case report of thrombocytopenia after the transfusion of a plasma unit containing HPA-5b antibodies [29]. This finding supports the notion that HPA-5b antibodies do not cause severe neonatal thrombocytopenia and thus that IVIg treatment is probably not necessary for pregnant women who are HPA-5b-immunized.

Although the association between HPA-1a immunization and HLA-DRB3*01:01 has been known for decades, the neonatal outcome has only recently been studied in HPA-1a-immunized women who are HLA-DRB3*01:01-positive and -negative, respectively. A systematic review and metanalysis was conducted by the International Collaboration for Transfusion Medicine Guidelines (ICTMG), which included four prospective and five retrospective studies [6]. In none of the four prospective studies (representing > 150,000 pregnant women) were there any HPA-1a-immunized, HLA-DRB3*01:01-negative women who gave birth to a severely thrombocytopenic child (platelet count < 50 × 10^9^/L). In the five retrospective studies, there were 13 severely thrombocytopenic newborns, of whom 2 suffered with ICH. These two cases were from a retrospective study that reported data from a large French reference laboratory over a 25-year period [30]. Clinical information about these two children with ICH was unfortunately not available. It is therefore not known if these two cases had concurrent neonatal or obstetric risk factors for ICH. Moreover, the grading of the bleeding was not reported; thus, it is not known if the ICH was of clinical importance or if it was an incidental finding, as ICH has been found in 26% of unselected asymptomatic newborns delivered by the vaginal route [31]. The metanalysis showed that the odds ratio for giving birth to a severely thrombocytopenic child, given the mother is HPA-1a-immunized, is only 0.08 if she is also HLA-DRB3*01:01-negative.

HLA-DRB3*01:01 typing is relevant for a pregnant woman who has been identified as HPA-1a-negative by virtue of being a blood donor or if she has a sister who previously had a child with FNAIT. The results from the systematic review and metanalysis conducted by ICTMG indicate that no special follow-up during pregnancy would be necessary. Even if she should develop HPA-1a antibodies during pregnancy, it is unlikely that the levels of HPA-1a antibodies will be high enough to produce significant thrombocytopenia in the fetus/newborn and treatment with IVIg should therefore not be necessary.

## 5. What Are the Risks of Not Treating Low-Risk Pregnancies with IVIg?

Based on the more recent data presented above [3,6,25], we suggest that pregnant women who are:HPA-1a-immunized and HLA-DRB3*01:01-negative;HPA-1a-immunized with a previous child with FNAIT but without ICH;HPA-5b-immunized;
should be categorized as “low-risk” pregnancies for having a fetus/child with ICH, and we would suggest that these women generally should not be treated with IVIg during pregnancy.

Critics may say that there still is a risk that low-risk women give birth to a child with ICH if IVIg is not administered during pregnancy, although this risk is low. There are not many cases in the literature that have described a low-risk woman not treated with IVIg who has had a child with ICH [32,33,34,35]. It is of course not possible to guarantee that a woman at low risk will not have a fetus/child with ICH, but the real question is whether this case could have been avoided if IVIg had been administered to the pregnant woman. IVIg treatment is also no guarantee for preventing ICH. There are several reports describing treatment failures [15,36,37]. IVIg treatment was initially introduced for high-risk pregnancies, i.e., cases where a woman previously had had a child with FNAIT-associated ICH. As these initial studies showed that the majority of these IVIg-treated women did not give birth to a child with ICH, it was assumed that IVIg treatment of low-risk pregnancies would also prevent ICH from happening. We question this assumption because there are no significant clinical or epidemiological data to support the usage of IVIg in low-risk pregnancies.

Although ICH is a known complication of FNAIT, Refsum et al. demonstrated that FNAIT only accounts for a minority of the neonatal clinical conditions which can cause ICH [28]. They analyzed 105 maternal samples obtained after searching a national register for neonates with ICH born after 32 weeks and found three platelet-immunized women. In addition to one woman with HPA-1a antibodies (1%), there were two women with HPA-5b antibodies (1.9%), which is similar to the percentage of HPA-5b-immunized women that would be expected by chance in a random population of pregnant women. Furthermore, in 194 consecutive ICH fetuses (FNAIT excluded), Coste et al. [38] screened for variants of the *COL4A1/COL4A2* genes, which have been associated with ICH, and identified pathogenic variants in 19%. Thus, fetal ICH is a highly heterogeneous condition and the probability of coincidence between fetal/neonatal ICH and maternal HPA antibodies is therefore high.

In the recent Norwegian study, none of the sixty-four women with low-risk pregnancies gave birth to a child with ICH, and fetal/neonatal ICH was only detected in two of seven high-risk pregnancies [25]. These data indicate that subsequent pregnancies of women who have had a fetus or newborn with FNAIT but without ICH usually have benign courses, and that the recurrence rate of FNAIT-associated ICH is lower than previously reported. Although some concerns have been raised regarding the Norwegian study [39], it provides the hitherto most reliable data regarding the efficacy of IVIg in low-risk pregnancies.

The natural history of pregnant HPA-5b-immunized women carrying an HPA-5b-incompatible fetus is not known. However, the clinical course of FNAIT in HPA-5b-immunized women has generally been considered as less severe than that of anti-HPA-1a-associated FNAIT. In addition, indirect evidence [29], as well as the recent study by Alm et al. [3], has questioned if HPA-5b antibodies can cause severe thrombocytopenia and argues that the association between neonatal ICH and HPA-5b-alloimmunisation is coincidental. Although ICH can occur in the fetus/child of an HPA-5b-immunized woman who is not treated with IVIg, a causal relationship with the presence of HPA-5b-antibodies is unlikely. The association could just be coincidental, similar to the case reported by de Vos and co-workers [27], where an HPA-5b-immunized mother without fetal–maternal incompatibility gave birth to a child with ICH.

As explained above, it is highly unlikely that a pregnant woman who is HPA-1a-immunized and HLA-DRB3*01:01-negative will have a fetus/child with ICH [6]. Thus, if an HLA-DRB3*01:01-negative mother gives birth to a child with ICH, this would more likely be coincidental than causal.

Since the study by Ernstsen et al. [25] is based on results from Norway, it could be questioned if these results can be extrapolated to other populations and ethnicities. However, despite this methodological weakness, the level of evidence for using IVIg treatment in low-risk pregnancies is much lower than the level of evidence for the opposite—that IVIg is not necessary in these pregnancies.

In the very rare scenario where an HPA-5b-immunized or an HPA-1a-immunized, HLA-DRB3*01:01-negative pregnant woman previously has had a child with FNAIT-suspected ICH, the counselling of the woman can be challenging. As such cases would be extremely rare, there are no data from the literature that could be used as guidance for the clinical management during pregnancy. The question in such rare cases is whether or not the ICH in the previous sibling was causally related to the mother’s alloantibodies; this is a question that rarely can be answered on an individual level. In this setting, the decision about prenatal IVIg treatment should be determined by the woman together with her physician.

## 6. Abstaining from IVIg Treatment for Low-Risk Pregnancies Would Significantly Reduce the Amount of IVIg Used for Women with Platelet Antibodies

The percentage of the three categories of low-risk pregnancies among pregnancies at risk of FNAIT in general can be calculated as follows:HPA-1a-immunized women with a previous child without ICH. Based on the data from Ernstsen and co-workers [25], 80% of all subsequent pregnancies of HPA-1a-immunized women belonged to the low-risk category (of a total of 474 women, 375 belonged to the low-risk group). Given that 80% of all FNAIT cases are associated with HPA-1a antibodies, approximately 64% (80% × 80%) will belong to the low-risk group.HPA-5b-immunized women. If 15% of FNAIT cases are associated with HPA-5b antibodies, and if we assume that all HPA-5b-immunized women belong to the low-risk category, there will be an additional 15% of all FNAIT cases that can be considered as low-risk pregnancies.HPA-1a-negative and HLA-DRB3*01:01-negative women. As these women only rarely become immunized, the majority of women belonging to this category will be identified by virtue of being HPA-1a-typed as a potential platelet donor or because they have a sister who has had a child with FNAIT. Hence, this group will be negligible compared to the other two groups of low-risk pregnancies.

If we changed the current clinical practice and stopped treating low-risk pregnancies with IVIg, we could reduce the use of IVIg in FNAIT by 79% (64% + 15%). This is important, first, because there is a worldwide shortage of IVIg [40]; secondly, because IVIg treatment is extremely costly (the IVIg used for treating one low-risk woman costs more than USD 150,000 or USD 300,000 depending on whether the dosage is 1 or 2 g/kg/week—see Table 1 for details); and thirdly, because the collection of plasma for the manufacture of IVIg requires tremendous donor efforts (>4 or 8 man-months for the collection of plasma for treating one low-risk woman, depending on the dosage of IVIg; Table 1).

As explained above, none of the 64 low-risk HPA-1a-immunized women gave birth to a child with ICH [25]. In addition, we have questioned the relevance of IVIg for HPA-5b-immunized women as well as HLA-DRB3*01:01-negative HPA 1a-immunized women. Consistent with the current programs of IVIg prophylaxis, all three categories of low-risk pregnancies would be treated with IVIg. Hence, according to a conservative assumption, it would be necessary to treat > 50 low-risk pregnancies to prevent one case of ICH. Thus, depending on the dose of IVIg, it would be necessary to use 76 or 152 kg IVIg for the prevention of one case of ICH, equivalent to USD 7,600,000 or USD 15,200,000 and 16.9 or 33.8 man-years in donor engagement (Table 1). We do not know of any other medical condition where such high costs are used for the prevention or cure of one case.

## 7. Conclusions

Here, we argue that the IVIg treatment of all pregnant women with a low risk of FNAIT results in significant overtreatment and with limited clinical benefit. Better tools for risk predictions, other than an individual’s FNAIT history, are urgently needed. The question of whether the risk of ICH in anti-HPA-1a-immunized women can be better predicted by in vitro methods, such as the identification of HPA-1a antibodies that bind to αVβ3 integrin [8] or afucosylated HPA-1a antibodies [47], must be addressed in a large prospective international collaborative trial. Finally, we suggest that the relevant scientific communities critically scrutinize available evidence for the efficacy of IVIg treatment of low-risk pregnancies and be open to revise current treatment guidelines for the prenatal management of FNAIT.

## Figures and Tables

**Table 1 jcm-12-05492-t001:** Cost of treating one woman with a low-risk pregnancy.

**Monetary costs**		
Body weight [41]	a	76 kg
No. of treatment weeks [42]	b	20
Dose of IVIg per week [14,42]	c	1–2 g/kg/week *
Total dose of IgG	d = a × b × c	1520–3040 g *
Price for IgG [43]	e	USD 100/g
Price for total dose of IgG	d × e	USD 152,000–304,000 *
**Donor engagement**		
Amount of plasma per plasmapheresis [44]	f	0.7 L
Amount of extractable IgG per L plasma [44,45]	g	5 g/L
Amount of plasma for treatment of one woman	h = d/g	304–608 L *
No. of apheresis procedures for treatment of one woman	i = h/f	869
Time for one apheresis procedure [46]	j	1.5 h
No. of apheresis hours for treatment of one woman	k = i × j	652–1303 h *
One man-month	l	80
No. of man-months for treatment of one woman	k/l	4–8 *

* The recommended dose used for treating low-risk pregnancies varies from 2 g of IVIg/kg/week in the US to 1 g/kg/week in most European countries. IVIg: intravenous immunoglobulin. IgG: immunoglobulin G.

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
