# Peer review of "Pregnant Women at Low Risk of Having a Child with Fetal and Neonatal Alloimmune Thrombocytopenia Do Not Require Treatment with Intravenous Immunoglobulin"

_jcm, 2023, doi:10.3390/jcm12175492_

Round 1

Reviewer 1 Report (Previous Reviewer 1)

Dear Authors,

After the first revision, the review looks fine to me now.

Author Response

Dear Authors,

After the first revision, the review looks fine to me now.

Authors’ reply: Thank you.

Reviewer 2 Report (New Reviewer)

Dear colleagues,

Thank you very much for your paper. I have some comments to make:

1. The authors are dealing with a very important topic.

2. The abstract ist very well written.

3. The entire review is very well structured and goes deep into the topic of FNAIT and IVIg. In addition, the references are appropriate.

4. Nevertheless, I think it would be beneficial to the reader if Section 4 ("Identification of less severe courses of FNAIT") was shortened.

Kind regards.

Author Response

Dear colleagues,

Thank you very much for your paper. I have some comments to make:

  1. The authors are dealing with a very important topic.
  2. The abstract ist very well written.
  3. The entire review is very well structured and goes deep into the topic of FNAIT and IVIg. In addition, the references are appropriate.
  4. Nevertheless, I think it would be beneficial to the reader if Section 4 ("Identification of less severe courses of FNAIT") was shortened.

Kind regards.

Authors’ reply: Thank you for your comments. Section 4 ("Identification of less severe courses of FNAIT") is the most important part of this review as we are explaining how to identify pregnant platelet-immunized women who will give birth to children no signs of FNAIT or less severe cases of FNAIT. Thus, it is difficult to make substantial cuts in this section without weakening our arguments. However, we have done our best to accommodate the wishes from this reviewer.  

Reviewer 3 Report (New Reviewer)

Dear 

This is a very interesting review. For a long time, we know about maternal IV immunoglobulin therapy for the treatment of FAINT. But, nowadays protocols introduce corticosteroids and IV immunoglobulins in the therapy of FAINT. (Am J Obstet Gyn 2021.).

I would like the authors to introduce us with some information about these protocols of maternal therapy. Administration of IV immunoglobulins and corticosteroids to the mother is indicated when we confirm in cases of paternal homozygote for HPA 1a. 

Sincerely 

Dear

I think that the article recommends Moderate editing of the English language required.

Author Response

Dear 

This is a very interesting review. For a long time, we know about maternal IV immunoglobulin therapy for the treatment of FAINT. But, nowadays protocols introduce corticosteroids and IV immuno­globulins in the therapy of FAINT. (Am J Obstet Gyn 2021.).

I would like the authors to introduce us with some information about these protocols of maternal therapy. Administration of IV immunoglobulins and corticosteroids to the mother is indicated when we confirm in cases of paternal homozygote for HPA 1a. 

Sincerely 

Authors’ reply: Many thanks for these comments. Our main message in this review is that the risk of neonatal ICH is very low in low-risk pregnancies. Therefore, treatment with IVIg is not warranted in this group of patients, and of this follows, that treatment with IVIg + corticosteroids is also not necessary. It is beyond the scope our review to discuss various treatment protocols for high-risk pregnancies, whether or not these protocols include corticosteroids in addition to IVIg. For these reasons, we have decided not to discuss these treatment protocols in this review.

Round 2

Reviewer 3 Report (New Reviewer)

Dear Authors,

Before Yours  explanation ( it must be better that included in Materijal and Methid Section),Your review could be accepted for publication.

Sincerely,

Ivana Babovic MD Ph.D Associated Professor

Faculty of Medicine, University of Belgrade

Serbia 

English language must have minor editing 

This manuscript is a resubmission of an earlier submission. The following is a list of the peer review reports and author responses from that submission.

Round 1

Reviewer 1 Report

This review focuses on several aspects highlighting the need of Intravenous Immunoglobulin (IVIG) for the treatment of Fetal and neonatal alloimmune thrombocytopenia (FNAIT) in pregnant women. The FNAIT management is complicated, and treatment depends on the severity of the condition and the risk to the newborn. The review addresses an important question whether IVIG should be used in low-risk pregnancy and the risks if IVIG is not administered. Overall, I found the review to be a well-written review with a lot of information. My comments are below.

A table summarizing the ‘Identification of less severe courses of FNAIT’ section could be a valuable addition to the review.

Authors should consider extending the paragraph 303-307 with more relevant information.

Authors should also include information regarding the factors affecting the FNAIT severity.

The section title ‘What are the consequences for the supply of plasma for IVIg production?’ could be more specific.

Reviewer 2 Report

   In the present review, the authors show that IVIg treatment may not be necessary for all pregnant women in order to reduce the risk of intracranial hemorrhage (ICH) in the fetuses and neonates.  This treatment may be omitted in pregnant women who are HPA-1a-immmunized and HLA-DRB3*01:01 negative, HPA-1a-immunized with a previous child with fetal and neonatal alloimmune thrombocytopenia but without ICH or HPA-5b-immunized, leading to the cut of costs and efforts. 

Major comments

1. Since the present conclusion is obtained by the data in Norway, authors should more precisely describe the ethnic difference. For example, alloimmune thrombocytopenia is correlated with the type of HPA and the of HLA.  Therefore, the present conclusion may not be generalized.

2. The title limited to Norwaigean  experience may be suitable. 

Minor comments

1. Font size is suddenly changed.

2. Description of "antibody" or "antibodies" is sometimes missing.  

 Please refer to minor comment #2 in the above section.

Reviewer 3 Report

Please see the file attached. Thanks

Minor editing of English language required. See the file for details.